# Transformers Know When They Don't Know: Layer-Wise Distance Awareness for OOD Detection

## Abstract

Out-of-distribution object detection (OOD-OD) is essential for building robust vision systems in safety-critical applications. While transformer-based architectures have become dominant in object detection, existing work on OOD-OD has primarily focused on OOD object synthesis or OOD detection scores, with limited understanding of the internal feature representations of transformers. In this work, we present the first in-depth analysis of transformer features for OOD-OD. Motivated by theoretical insights that *input distance awareness* – the ability of feature representations to reflect the distance from the training distribution – is a key property for predictive uncertainty estimation and reliable OOD detection, we systematically evaluate this property across transformer layers. Our analysis reveals that certain transformer layers exhibit heightened input distance awareness. Leveraging this observation, we develop simple yet effective OOD detection methods based on features from these layers, achieving state-of-the-art performance across multiple OOD-OD benchmarks. Our findings provide new insights into the role of transformer representations in OOD detection. **Code and additional experiments are in the Supp.**

## 1 Introduction

Table 1: **We conduct the first study to systematically analyze transformer features for OOD detection.** Most existing methods focus on CNN-based object detectors, and many techniques are tailored specifically to CNN models and backbones, e.g., SAFE (Wilson et al., 2023). Meanwhile, recent efforts have focused on techniques for synthesizing OOD objects and OOD detection scores. In contrast, feature representations for OOD detection in transformer-based object detectors remains largely unexplored.

| | Arch. | | Research focus | | |
|---|---|---|---|---|---|
| | CNN | **Transformer** | OOD Synthesis | OOD score | **Feature** for OOD |
| VOS (Du et al., 2022b) | ✓ | | ✓ | | |
| FFS (Kumar et al., 2023) | ✓ | | ✓ | | |
| SR-VAE (Wu & Deng, 2023) | ✓ | | ✓ | | |
| DFDD (Wu et al., 2023) | ✓ | | ✓ | | |
| MPD (Aming & Deng, 2024) | ✓ | | ✓ | | |
| SIREN (Du et al., 2022a) | | ✓ | | ✓ | |
| VisTa (Zhang et al., 2025b) | ✓ | | | ✓ | |
| SAFE (Wilson et al., 2023) | ✓ | | ✓ | | ✓(CNN) |
| SyncOOD (Liu et al., 2024) | ✓ | | ✓ | | |
| **Ours** | | ✓ | | | ✓(Transformer) |

Object detection is one of the most critical tasks in computer vision. Currently, state-of-the-art (SOTA) object detectors (Zhao et al., 2024a;b; Hou et al., 2024) are trained on closed-set datasets,

which can lead to overconfident predictions on outlier samples (Dhamija et al., 2020; Nguyen et al., 2015). In real-world deployments, such as autonomous driving, unknown objects often emerge, and failing to detect them can result in serious accidents (Nitsch et al., 2021). As a result, the research community is actively pursuing out-of-distribution detection in both image classification (Wang & Li, 2024; Tang et al., 2024; Ming et al., 2022; Yuan et al., 2024; Xue et al., 2024; Bai et al., 2024; Zhang et al., 2024) and object detection (Liu et al., 2024; Wilson et al., 2023; Kumar et al., 2023; Wu & Deng, 2023; Wu et al., 2023; Aming & Deng, 2024; Du et al., 2022a;b) to better recognize outlier samples and enhance the trustworthiness of model predictions.

**Research gaps in OOD-OD.** Recent OOD-OD approaches commonly operate by extracting features from one or several layers of a pretrained detector and then applying techniques such as energy-based scoring or lightweight classifiers (e.g., MLPs) to distinguish in-distribution (ID) and out-of-distribution (OOD) samples. While effective to some extent, this line of work still leaves several critical gaps, as summarized in Table 1. First, although transformer-based detectors such as MS-DETR (Zhao et al., 2024a), ViTDET (Li et al., 2022) have become popular in modern object detection, most OOD-OD methods remain focusing on CNN-based backbones like Faster-RCNN. For example, VOS (Du et al., 2022b), FFS (Kumar et al., 2023), DFDD (Wu et al., 2023), and SAFE (Wilson et al., 2023) are all developed for CNN backbones. Only SIREN (Du et al., 2022a) leverages a transformer-based model, but they treats transformers as monolithic units, without investigating which specific internal layers or components are most sensitive to OOD signals. Second, the majority of existing methods extract features exclusively from the final layer of the detector, assuming that high-level representations are sufficient for capturing distributional shifts. However, this overlooks the representational diversity encoded across intermediate layers, which may offer more robust cues for OOD detection, especially in deep transformer-based models (Zhang et al., 2022; Sonkar & Baraniuk, 2023). As a result, it remains unclear how to best utilize the internal structure of transformers to enhance OOD-OD performance.

**In this work**, we conduct the first study to analyze transformer layers for OOD object detection and propose a simple and effective method based on our analysis of transformer layer characteristics. Our work builds on the theoretical foundation of *input distance awareness*. Particularly, Liu et al. (2020a) identify input distance awareness as a necessary condition for reliable uncertainty estimation in deep networks. Their formulation emphasizes that this property depends on a bi-Lipschitz mapping between input space and hidden representations, ensuring that distances in the feature space reflect meaningful differences in the input distribution. Motivated by this insight, we hypothesize that certain internal layers in transformer-based detectors better preserve input distance, and thus are more effective for OOD detection. We propose a framework for layer-wise sensitivity analysis to quantify this distance-preserving property. Based on our analysis, we propose a simple and effective method to identify and aggregate features from the most sensitive layers. This principled approach yields a simple, architecture-agnostic method that consistently improves OOD detection performance without retraining or architectural modifications. Focusing on OOD-OD, our contributions can be summarized as follows. **Firstly**, we bridge a critical gap by proposing a framework to analyze internal layers of Transformer architectures for OOD-OD effectiveness, going beyond the conventional focus on penultimate layer or CNN-based features. **Secondly**, we are the first to exploit theoretical results of input distance awareness in Transformer architectures for OOD-OD, leveraging Lipschitz analysis to quantify sensitivity. **Thirdly**, we achieve SOTA performance on challenging OOD-OD benchmarks without retraining the object detector. Our results are demonstrated across ID-OOD dataset setups using two important Transformer-based object detectors. **Finally**, while previous approaches rely on highly specialized OOD detection methods tied to specific object detectors, our approach is model-agnostic. It is based solely on extracted features and does not require architectural modifications, making it more easily applicable for new Transformer-based object detectors.

## 2 RELATED WORKS

**OOD Dection for Image Classification** can be broadly categorized into fine-tuning-based and post-hoc approaches. Fine-tuning-based mitigate overconfidence on OOD samples by introducing random noise, shuffling image patches (Lee et al., 2017), using auxiliary datasets (Hein et al., 2019), or synthesizing outliers (Du et al., 2022b; Tao et al., 2023), though their performance depends on outlier quality and may degrade ID accuracy. Post-hoc methods require no retraining; MSP (Hendrycks &

Gimpel, 2016) inspired variants such as ODIN (Liang et al., 2017), Energy Score (Liu et al., 2020b), ReAct (Sun et al., 2021), and DICE (Sun & Li, 2022). Other works exploit feature-space distances, e.g., Mahalanobis (Lee et al., 2018) or k-NN (Sun et al., 2022). Recent studies (Tang et al., 2024) analyzes different layers, moving beyond methods that rely solely on logits (Lee et al., 2018) or penultimate layer features (Sun et al., 2022), emphase explore of intermidate layers representation for OOD detection. Highlighting the importance of exploring intermediate-layer representations for OOD detection.In this work, we are the first to analyze the sensitivity of different intermediate layers in object detectors, without being limited to a specific architectural variant, and to propose a sensitivity-guided selection criterion for identifying the most effective layers from which to extract object-specific features for OOD detection.

**OOD Detection for Object Detection.** Early approaches focused on generating synthetic OOD data, such as VOS (Du et al., 2022b), NPOS (Tao et al., 2023), DFDD (Wu et al., 2023), SRVAE (Wu & Deng, 2023), and FFS (Kumar et al., 2023). In contrast to these synthesis-based approaches, SIREN (Du et al., 2022a) does not require any OOD samples; instead, it introduces an auxiliary model to reshape ID feature representations. RUNA (Zhang et al., 2025a) addresses the cognitive limitations of object detectors by integrating CLIP (Radford et al., 2021) into the detection pipeline, performing multi-step OOD detection through repeated CLIP-based image encoding. Unlike VOS, NPOS, SIREN, DFDD, SR-VAE, and FFS, SAFE (Wilson et al., 2023) follows a post-hoc detection paradigm. It proposes a feature selection mechanism that identifies sensitivity-aware representations and trains an MLP for OOD detection. However, SAFE is restricted to specific CNN components, namely batch normalization and skip connections. Our approach departs from these constraints by proposing a Bi-Lipschitz-based sensitivity analysis to identify distance-aware intermediate layers, without relying on any architectural assumptions. This allows our method to generalize effectively across a wide range of object detectors.

**Uncertainty estimation**. Traditional approaches to uncertainty estimation, such as deep ensembles and Bayesian neural networks, are computationally expensive. To address this, SNGP (Liu et al., 2020a) explores distance-awareness as a means to measure the distributional shift between test and training samples, thereby supporting uncertainty estimation in deterministic models. Since deep neural networks are not inherently designed to preserve input distance sensitivity, numerous studies have introduced regularization techniques to promote this property. These studies include sensitivity-aware training (Liu et al., 2020a) and spectral normalization (Miyato et al., 2018). Architectural factors such as residual connections have also been investigated for their influence on sensitivity (Mukhoti et al., 2021). Several of these works (Liu et al., 2020a; Van Amersfoort et al., 2020; Mukhoti et al., 2021) emphasize the importance of distance-awareness properties for OOD detection.

Rather than focusing solely on the penultimate layer and retrained the model, as in Liu et al. (2020a), we analyze the sensitivity of each layer in pretrained object detectors. We further propose variations of the sensitivity formulation to identify the most sensitive layers for OOD detection.

## 3 PRELIMINARIES

We start with a pretrained object detector $f$, which takes an input image $x$ and outputs $D$ object predictions - each with a class label and a bounding box. However, some predicted objects may be OOD, despite being assigned high confidence by the model. The goal of OOD-OD is to classify each predicted object as either ID or OOD, thereby improving the reliability of object detectors in real-world deployments.

### 3.1 INPUT DISTANCE AWARENESS

A reliable measure of uncertainty, particularly for detecting OOD inputs, requires a deterministic model to be input distance-aware, meaning it can quantify how far a test example lies from the training data manifold. Without such awareness, models often produce overconfident predictions for OOD inputs, even when these inputs are far from the known data distribution. This issue arises because uncertainty is frequently associated with the decision boundary, which is learned through the model's task-specific optimization, rather than with the true distance from the training data manifold (Liu et al., 2020a).

Consider a deep neural network with logits defined as:

$$\text{logit}(x) = g \circ h(x),$$

where $h : \mathcal{X} \to \mathcal{H}$ is the hidden mapping that transforms input $x$ into a feature representation $h(x)$, and $g$ maps $h(x)$ to class logits. Following Liu et al. (2020a), *input distance awareness* requires two conditions:

- **Distance-aware output layer** ($g$): The output function must produce uncertainty estimates that reflect hidden-space distances $\|h(x) - h(x')\|_{\mathcal{H}}$, for example, in Gaussian process-based output layers.
- **Distance-preserving hidden mapping** ($h$): Distances in the hidden space meaningfully correspond to distances in the input space $\|x - x'\|_{\mathcal{X}}$.

When both conditions hold, the model's uncertainty estimates naturally scale with the distance from the training domain, improving both calibration and OOD detection. In our work, we focus on the distance-preserving of the hidden layers. In Liu et al. (2020a), they improve distance preservation in intermediate layers using spectral normalization and analyze it using the Bi-Lipschitz equation:

$$K_1 \|x - x^*\|_I \leq \|h(x) - h(x^*)\|_F \leq K_2 \|x - x^*\|_I \tag{1}$$

Here, $x$ and $x^*$ are two distinct inputs, while $\|\cdot\|_I$ and $\|\cdot\|_F$ denote distance metrics in input and feature spaces, respectively. $K_1$ and $K_2$ are constants. Importantly, the lower Lipschitz bound $K_1\|x - x^*\|_I \leq \|h(x) - h(x^*)\|_F$ characterizes **sensitivity** of the hidden representation, ensuring their effectiveness in preserving meaningful changes in the input manifold, which is important for OOD detection.

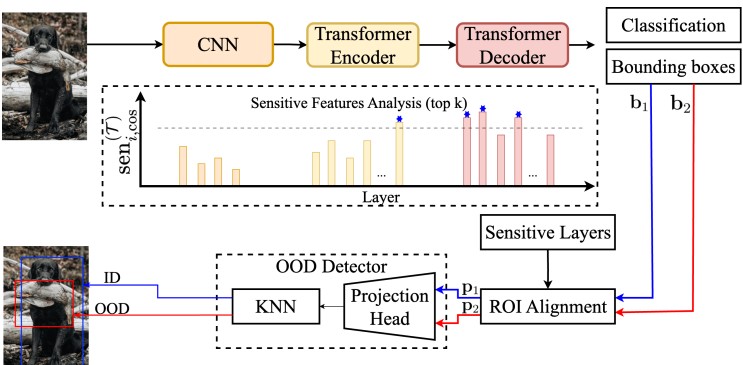

Figure 1: Our propose OOD-OD method, SeFea, which leverage sensitive object-specific features (OSFs) in transformer-based object detector. The top $k$ most sensitive transformer layers are pre-identified using $\text{sen}_{i,\cos}^{(\mathcal{T})}$ sensitivity metric. Feature maps from these layers are extracted and processed through ROI Align, then concatenated to form the OSFs, denoted as $p_d$. These OSFs are then passed through the OOD Detector module, which outputs OOD predictions for each detected object, enabling discrimination between in-distribution (blue) and out-of-distribution (red) objects.

## 4 PROPOSED METHOD

Motivated by prior theoretical work on sensitivity and the need for a fine-grained, layer-wise understanding of OOD detection, we pose a central question: *How can sensitivity be leveraged to identify transformer layers most effective for OOD-OD?* In what follows, we first present an overview of our method, which utilizes sensitive transformer layers for OOD-OD. We then introduce our framework for identifying these sensitive transformer layers, grounded in the lower Lipschitz bound formulation in Eq. 1.

### 4.1 OVERVIEW OF PROPOSED OOD-OD METHOD: SEFEA

Figure 1 outlines our sensitive transformer feature-based OOD-OD method, SeFea. Transformer layers are first ranked using a sensitivity-based algorithm to characterize input–distance awareness.

The detailed design of this algorithm are described in a later section. Based on the resulting sensitivity values, we select the most sensitive layers for OOD detection. Feature extraction is performed by using a tracker to collect the corresponding feature maps from the object detector. To obtain object-specific features (OSFs), we use the predicted bounding boxes $\{b_1, \ldots, b_D\}$ to crop the object regions of each feature map. For each object $d$ and each layer $l$, we extract features denoted as $O_{l,d}$. Each $O_{l,d}$ is then spatially pooled to a size of $1 \times 1$ (width $\times$ height) to produce a feature vector $p_{l,d}$, whose dimensionality matches that of the corresponding feature map $l$. The final OSF vector for each object, $p_d$, is formed by concatenating the feature vectors $\{p_{1,d}, \ldots, p_{k,d}\}$ from all sensitive layers, which form our sensitive feature (SeFea).

Afterward, the OSFs are passed through the OOD Detector module for OOD detection. Specifically, we explore the detection head proposed in Du et al. (2022a), which consists of a projection head implemented as a sequence of fully connected layers. This head is trained to enforce a von Mises-Fisher distribution on the OSF embeddings. The OOD score is then computed using the KNN distance in this compact and normalized feature space.

Our approach is architecture-agnostic and does not rely on the common assumption that the penultimate-layer representation is well-suited for OOD detection. Instead, it systematically selects the most sensitive layers, making it applicable to a wide range of object detectors. Unlike SAFE (Wilson et al., 2023), which imposes constraints such as the presence of batch normalization and skip connections, or methods that depend solely on penultimate-layer features (Du et al., 2022a;b; Kumar et al., 2023), our method can be applied without such architectural restrictions.

## 4.2 SENSITIVE LAYERS ANALYSIS

A key question is how to rank the transformer layers for OOD detection. Motivated by prior theoretical analyses of input distance awareness in hidden-layer representations, we design *layer-wise sensitivity metrics* to quantify how strongly each transformer layer responds to changes in the input. Following Liu et al. (2020a), transformer layers with higher sensitivity are expected to be more effective for OOD detection.

To achieve more accurate comparisons across layers with different feature dimensionalities, we explore *dimension-invariant similarity measures*. Normalized Euclidean distance achieves dimension invariance by normalization with dimensionality, yielding the average per-dimension difference. Cosine similarity is dimension-invariant because it measures only the angle between two vectors, normalizing their magnitudes and disregarding the dimensionality of the feature space. We explore these metrics for more accurate sensitivity comparisons across transformer layers of varying dimensionalities, mitigating potential bias toward higher-dimensional layers.

Using the Bi-Lipschitz's lower bound in Eq. 1, and applying normalization with respect to both the input and feature dimensionalities, we develop the following equation for quantifying sensitivity awareness for the $i$-th transformer layer:

$$\text{sen}_{i,\text{Euc}}^{(\mathcal{T})} = \frac{C_x}{n_{\text{pairs}} \cdot C_i} \sum_{j=1}^{n_{\text{pairs}}} \frac{\|f_i(x_j^{(\mathcal{T})}) - f_i(x_j)\|}{\|x_j^{(\mathcal{T})} - x_j\|}. \tag{2}$$

Beside the normalized Euclidean distance, we further consider cosine similarity to develop the following equation for quantifying sensitivity for the $i$-th transformer layer:

$$\text{sen}_{i,\text{cos}}^{(\mathcal{T})} = \frac{1}{n_{\text{pairs}}} \sum_{j=1}^{n_{\text{pairs}}} \frac{1 - \cos\big(f_i(x_j^{(\mathcal{T})}), f_i(x_j)\big)}{1 - \cos\big(x_j^{(\mathcal{T})}, x_j\big)}. \tag{3}$$

We explore several types of transformation $\mathcal{T}$ to obtain $x_j^{(\mathcal{T})}$ from $x_j$ for the distance computation.

- Random Sample: $x_j^{(\mathcal{T})}$ is another random sample different from $x_j$.

- FGSM: $x_j^{(\mathcal{T})} = FGSM(x_j)$ denoting adversarially perturbed sample obtained via Fast Gradient Sign Method (FGSM) (Goodfellow et al., 2014), is used to induce perturbations on $x_j$ to obtain $x_j^{(\mathcal{T})}$.

- Gaussian: $x_j^{(\mathcal{T})} = x_j + \mathcal{N}(\mu, \sigma^2)$, where $\mathcal{N}(\mu, \sigma^2)$ is Gaussian noise with pre-defined mean and standard deviation.

In Eq. 2 and Eq. 3, $C_x$ denotes the dimensionality of the input space, and $C_i$ denotes the dimensionality of the feature of the $i$-th transformer layer. $n_{\text{pairs}}$ represents the predefined number of randomly sampled pairs used in the sensitivity computation. The sensitivity is computed for each pair, and the mean over all pairs is then taken to obtain the final sensitivity score.

In our OOD-OD detector, instead of choosing features solely from the highest-sensitivity layer for OOD detection, we also aggregate features from several high-sensitivity-aware layers. Neural networks learn hierarchical representations: lower layers capture basic features, while higher layers encode more complex concepts. By concatenating features from various layers, our OSFs harness a richer spectrum of information, enhancing their ability to distinguish ID from OOD detections. This layer-feature integration aligns with the principle that combining diverse representation levels can improve the robustness and accuracy of OOD detection systems. The number of high-sensitivity layers used for concatenation is explored in the supplementary material.

### 4.2.1 SENSITIVITY & OOD PERFORMANCE CORRELATION

Table 2: **Validation of effectiveness of our proposed sensitivity metrics in selecting effective transformer layers for OOD-OD.** Pearson correlation between layer-wise sensitivity scores (Eq. 2 and Eq. 3) and OOD detection performance (AUROC) across all layers of MS-DETR and ViTDET object detectors. Our analysis shows that Cosine distance paired with Random Sampling yields the highest correlation, indicating its effectiveness in identifying effective transformer layer for OOD-OD.

| Model | Sensitivity Calculation | Transformation Type ($\mathcal{T}$) | Pearson↑ | | | |
|---|---|---|---|---|---|---|
| | | | VOC | | BDD | |
| | | | MS-COCO | OpenImages | MS-COCO | OpenImages |
| MS-DETR | Euclidean | Random Sample | 0.397 | 0.393 | 0.366 | 0.376 |
| | | FGSM | 0.286 | 0.256 | 0.224 | 0.239 |
| | | $\mathcal{N}(10,30)$ | 0.324 | 0.295 | 0.260 | 0.274 |
| | | $\mathcal{N}(10,150)$ | 0.412 | 0.388 | 0.292 | 0.305 |
| | Cosine | Random Sample | **0.612** | **0.608** | 0.677 | 0.653 |
| | | FGSM | 0.363 | 0.344 | 0.390 | 0.387 |
| | | $\mathcal{N}(10,30)$ | 0.471 | 0.465 | 0.540 | 0.528 |
| | | $\mathcal{N}(10,150)$ | 0.592 | 0.586 | **0.687** | **0.661** |
| ViTDET | Euclidean | Random Sample | 0.248 | 0.244 | 0.148 | 0.184 |
| | Cosine | Random Sample | **0.816** | **0.775** | **0.697** | **0.690** |

To validate effectiveness of Eq. 2 and Eq. 3 in selecting transformer layers for our OOD detector, we analyze their correlation with OOD detection accuracy. The details of transformer-based models (MS-DETR, ViTDET), and the details of the implementation of the sensitivity calculation, such as $n_{\text{pairs}}$ and the dimension of the input space, are provided in the Experiment section and Supp. The correlation is measured using the *Pearson* correlation coefficient.

This sensitivity–OOD accuracy correlations are reported in Table 2. We observe that cosine similarity consistently achieves higher correlations with OOD detection accuracy than Euclidean distance across both MS-DETR and ViTDET, with the gap being particularly pronounced for ViTDET. Among different types of transformation $\mathcal{T}$, Random Sample transformation generally yields the strongest correlations, while FGSM perturbations produce noticeably lower values. Gaussian noise perturbations—especially with higher variance—often match or slightly surpass the performance of normal pairs. These findings indicate that angular-based similarity measures with Random Sample transformation is effective in identifying effective transformer layer. Therefore, in our method, we rank layers by Eq. 3 to compute $\text{sen}_{i,\cos}^{(\text{RandomSample})}$, and the OSFs used in our method are collected from the $k$ most sensitive layers according to this ranking (See Figure 1).

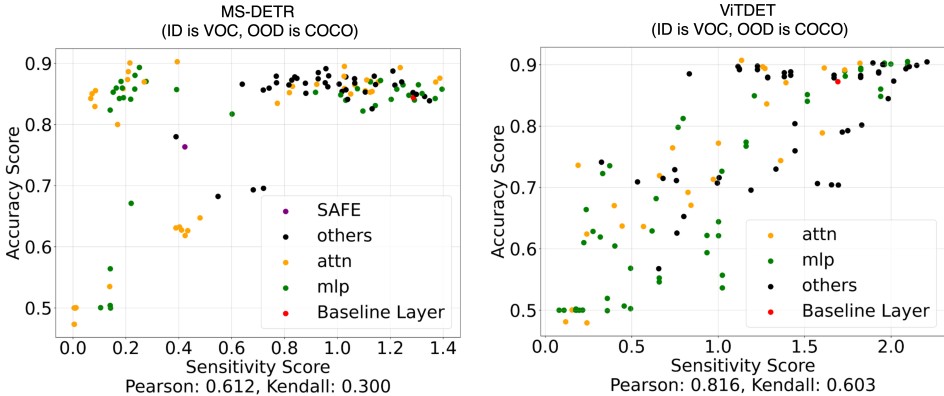

Figure 2: Correlation between sensitivity and OOD detection performance across all layers. The layers considered include SAFE (Wilson et al., 2023), penultimate (Baseline Layer), attention layers (attn), MLP layers (mlp), and other components (others). Each point represents an individual layer, with its sensitivity score $\text{sen}_{i,\cos}^{(\mathcal{T})}$ and the corresponding OOD detection performance (AUROC) based on the OSFs extracted from that layer. The SIREN-KNN is used as the OOD detector, and sensitivity is computed using inputs obtained from Random Sample transformation.

Figure 2 visualizes sensitivity–OOD performance relationship across different layers. We find a clear trend where layers with higher sensitivity tend to exhibit higher OOD detection accuracy. Notably, many intermediate transformer layers show higher sensitivity and OOD accuracy than the penultimate layer, suggesting that the richest OOD cues often reside in intermediate layers. Overall, there exists noticeable correlation between OOD accuracy and sensitivity, and our sensitivity metric serves as a viable selection signal for identifying sensitive layers.

Table 3: Comparison of the proposed method SeFea (*Ours*) with existing OOD-OD methods (MSP (Hendrycks & Gimpel, 2016), SAFE (Wilson et al., 2023) and SIREN (Du et al., 2022a)) on two transformer-based architectures: MS-DETR and ViTDET. All methods use **SIREN-KNN** (left) or **SIREN-vMF** (right) as the OOD detector, except for MSP. Evaluation is performed on PASCAL-VOC as the ID dataset, and MS-COCO and OpenImages as the OOD datasets. Performance is reported using AUROC and FPR95 metrics.

| | | SIREN-KNN | | | | SIREN-vMF | | | |
|---|---|---|---|---|---|---|---|---|---|
| | | ID: PASCAL-VOC | | ID: BDD | | ID: PASCAL-VOC | | ID: BDD | |
| | Method | OOD: MS-COCO/OpenImages | | | | OOD: MS-COCO/OpenImages | | | |
| | | AUROC↑ | FPR95↓ | AUROC↑ | FPR95↓ | AUROC↑ | FPR95↓ | AUROC↑ | FPR95↓ |
| **MS-DETR** | MSP | 77.37/69.48 | 68.55/75.08 | 79.22/82.28 | 80.26/81.30 | 77.37/69.48 | 68.55/75.08 | 79.22/82.28 | 80.26/81.30 |
| | SAFE | 77.26/79.29 | 78.54/75.12 | 82.90/80.68 | 72.50/75.08 | 76.08/84.33 | 76.20/60.63 | 72.78/75.19 | 76.15/75.29 |
| | SIREN | 84.39/80.75 | 53.47/59.37 | 88.60/89.75 | 64.73/ 61.38 | 73.14/69.90 | 81.50/82.68 | 76.48/78.09 | 80.32/82.01 |
| | *Ours* | **86.38/86.22** | **50.71/50.13** | **88.95/90.16** | **61.57/60.37** | **84.37/84.47** | **58.05/55.83** | **88.17/87.39** | **63.02/60.69** |
| **ViTDET** | MSP | 73.75/73.31 | 87.13/86.65 | 71.41/72.92 | 87.50/86.75 | 73.75/73.31 | 87.13/86.65 | 71.41/72.92 | 87.50/86.75 |
| | SIREN | 87.24/86.80 | 55.71/52.56 | 87.17/87.62 | 62.98/62.15 | 83.67/**87.34** | 58.05/**50.72** | 74.44/74.19 | 72.66/72.01 |
| | *Ours* | **90.41/90.79** | **40.13/41.52** | **89.60/91.28** | **46.90/44.14** | **84.08**/85.49 | **54.69**/50.88 | **90.02/93.94** | **42.10/30.26** |

# 5 EXPERIMENTS

## 5.1 EXPERIMENTAL SETUP

**Datasets** We use the same ID and OOD datasets following Du et al. (2022b) for all our experimental setups. Specifically, PASCAL-VOC (VOC) and Berkeley DeepDrive-100K (BDD) serve as the ID datasets, while subsets of MS-COCO and OpenImages function as the OOD datasets. The OOD sets are curated to ensure the absence of any classes present in the ID sets.

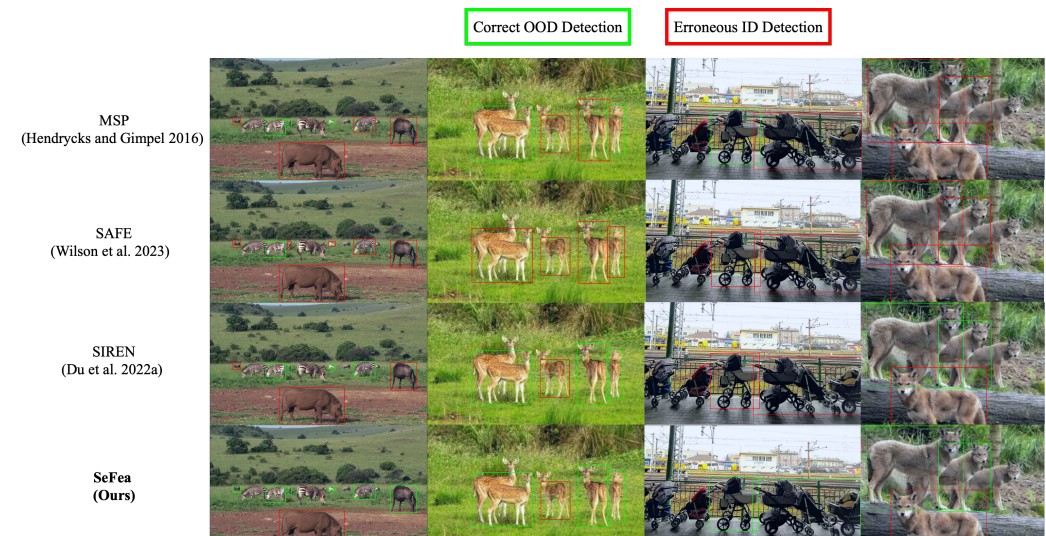

Figure 3: Qualitative visualization of OOD-OD detection results from the MS-DETR object detector with Pascal-VOC as the ID dataset. Detection results are obtained by comparing the OOD score against the threshold at FPR95. Green boxes indicate that the OOD detector correctly classifies the object as OOD, while red boxes denote misclassification as ID. All images are chosen such that none of the 20 Pascal-VOC classes appear in the scenes.

**Evaluation Metrics** AUROC and FPR95 are used, with details provided in the appendix. We follow the evaluation protocol of Wilson et al. (2023); Du et al. (2022b), where AUROC and FPR95 are computed after filtering out low-confidence bounding box predictions.

**Object detectors** Most prior OOD detection methods have been developed for the Faster R-CNN architecture, whereas our work focuses on transformer-based architectures—specifically MS-DETR and ViTDET—which, to the best of our knowledge, have not been previously explored for OOD-OD. Consequently, we re-implement several SOTA methods, including MSP (Hendrycks & Gimpel, 2016), SAFE (Wilson et al., 2023), and SIREN (Du et al., 2022a), within our MS-DETR and ViT-DET framework to ensure a fair comparison.

## 5.2 IMPLEMENTATION

**Network Architecture** To analyze and identify sensitive layers in transformer-based architectures for OOD-OD, we utilize MS-DETR (Zhao et al., 2024a) and ViTDET (Li et al., 2022), paired with ResNet-50(He et al., 2016) and ViT-B(Dosovitskiy et al., 2021) backbones, respectively. Since these architectures are originally trained on the COCO dataset, we retrain them on the designated ID datasets prior to evaluation. The pre-trained models are ensured to achieve strong detection performance on the ID validation sets, and provide a fair comparison of OOD detection given the predicted bounding boxes. Details of the pre-training results on the ID datasets are provided in the supplementary material. It is important to note that although SIREN (Du et al., 2022a) retrains both the object detector and the OOD detector using a customized loss function, in our implementation, we only train the SIREN OOD detector while keeping the pretrained object detector frozen.

**Feature Extraction** The overall pipeline is depicted in Figure 1; Transformer features are identified using Eq. 3, which has been validated to be effective in the previous section. More details can be found in the supplementary material. **OOD Detectors** We adopt the SIREN framework (Du et al., 2022a) with two types of prototype-based detection heads. The extracted OSFs are first passed through a projection layer to obtain modulated features tailored for the OOD detection task. In the SIREN-KNN setting, class prototypes are constructed via KNN clustering. In contrast, the SIREN-vMF setting models class-wise prototypes using von Mises-Fisher (vMF) distributions. OOD samples are identified as those with low probability under all class prototypes.

## 5.3 QUANTITATIVE ANALYSIS

Tables 3 compare our method with existing SOTA OOD-OD approaches with two types of OOD detector:

SIREN-KNN. Across both transformer-based architectures (MS-DETR and ViTDET) and all ID/OOD dataset configurations, SeFea consistently outperforms MSP, SAFE, and SIREN baselines, achieving the highest AUROC and lowest FPR95 scores. Compared to the strongest baseline (SIREN), AUROC improvements range from +0.35% to +5.47%, while FPR95 reductions are often substantial—frequently exceeding 10%. Notably, ViTDET benefits more from SeFea than MS-DETR, particularly in FPR95 (e.g., reducing from 55.71% to 40.13% on VOC/COCO and from 62.98% to 46.90% on BDD/COCO), suggesting that our sensitivity-guided intermediate-layer selection (using Eq. 3) outperforms the penultimate-layer feature used in SIREN (Du et al., 2022a), or batch norm/skip connection features used in SAFE (Wilson et al., 2023). These results highlight the effectiveness of SeFea in improving OOD robustness and demonstrate its architecture-agnostic applicability through a simple sensitivity-based heuristic.

SIREN-vMF. When using the SIREN-vMF OOD scoring function, SeFea again demonstrates consistent improvements across nearly all configurations, except for one case, ranking second on ViT-DET with VOC/OpenImages setup, with only a small gap of 1.85% in AUROC and 0.16% in FPR95. Notably, for the more challenging BDD as the ID setting, the FPR95 improves significantly, dropping from 72.01% to 30.26% on the BDD/OpenImages setup, highlighting SeFea's strong capability to reduce false positives in certain scenarios. While improvements are also observed for MS-DETR, the relative gains are smaller than those on ViTDET.

## 5.4 QUALITATIVE ANALYSIS

Figure 3 provides qualitative results. It presents OOD detection results for MSP, SAFE, SIREN, and our SeFea method (from top to bottom) on a set of OOD images, with all visualizations based on the SIREN-KNN detector. MS-DETR is used as the detector, Pascal-VOC serves as the ID dataset, and COCO is used as OOD to determine the FPR95 threshold. Across all example scenes, SeFea consistently produces more correctly classified OOD objects (green boxes) and fewer false-positive ID predictions (red boxes) than the SIREN baseline. This improvement is especially pronounced in the first column, where SIREN misses several OOD objects, while SeFea correctly flags nearly all deer as OOD. In the wolf example (last column), MSP and SAFE misclassify multiple wolves as ID, whereas SeFea substantially reduces these errors. This indicates stronger robustness when OOD objects share strong visual similarities with ID classes. Moreover, SeFea demonstrates consistent detection capability across varying object scales, from small to large. These results provide the first systematic evidence that sensitivity-guided intermediate-layer selection—rather than defaulting to the penultimate layer—offers a principled, architecture-independent path to SOTA OOD detection in object detection.

**Additional results and ablation are included in Appendix.**

## 6 CONCLUSION

We address the overlooked research gap of understanding intermediate-layer representations for OOD detection from a sensitivity perspective, challenging the common assumption that the penultimate layer is always optimal. Our experiments show that intermediate layers often encode richer and more informative cues for distinguishing ID from OOD objects. By quantifying input–distance awareness via a sensitivity metric, we find a strong correlation with OOD detection performance—particularly in ViTDET—and demonstrate that sensitivity serves as an effective criterion for layer selection. Extending the analysis across multiple detector architectures and diverse ID/OOD setups further validates the robustness of this finding. Our method is architecture-agnostic, as it does not depend on specialized layers, making it broadly applicable across object detection pipelines. We remark that our work validates the theoretical results of Liu et al. (2020a) on input distance awareness in practical transformer-based object detection models.

**Limitation.** Our work studies two important transformer-based object detection models due to computation constraint. Including more transformer-based detectors can further strengthen our findings.

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

## A APPENDIX

In this supplementary material, we provide additional ablation studies, more qualitative visualizations, inference time measurements, object detector accuracies on the ID datasets, and the algorithm for easier understanding. These details are not included in the main paper due to space limitations.

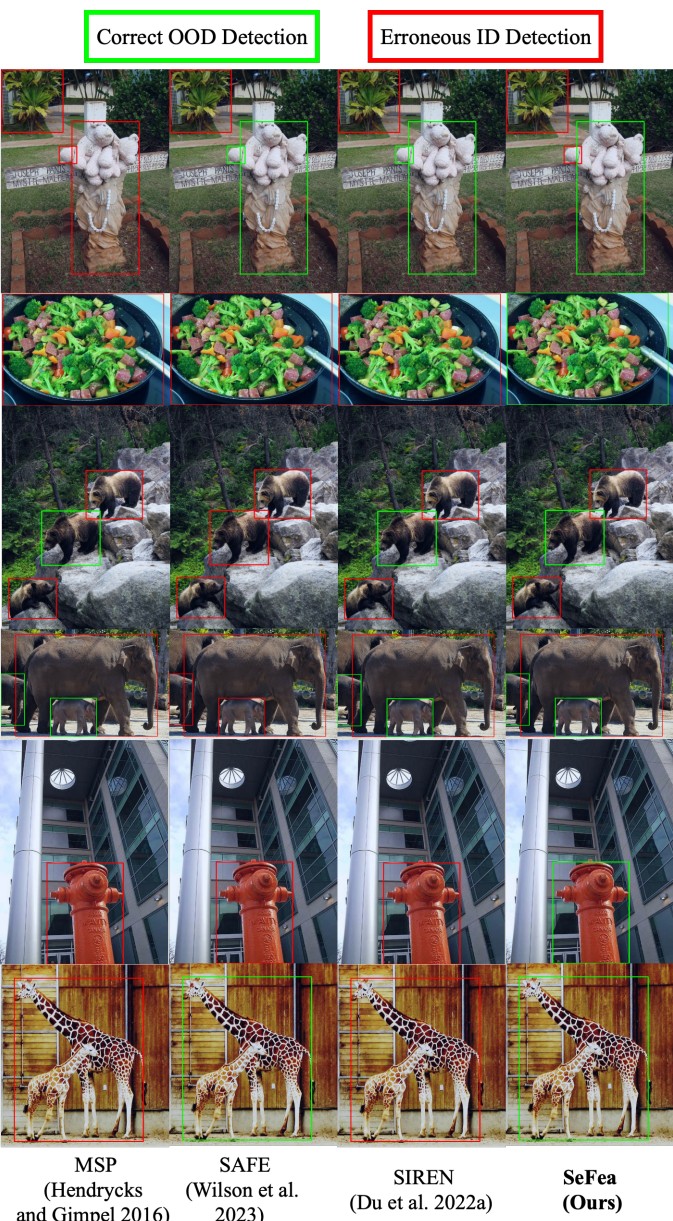

Figure 4: Qualitative visualization of OOD-OD detection results from the MS-DETR object detector with **Pascal-VOC as the ID dataset**, the images is collected from the COCO dataset. Detection results are obtained by comparing the OOD score against the threshold at FPR95. Green boxes indicate that the OOD detector correctly classifies the object as OOD, while red boxes denote misclassification as ID. All images are chosen such that none of the 20 Pascal-VOC classes appear in the scenes.

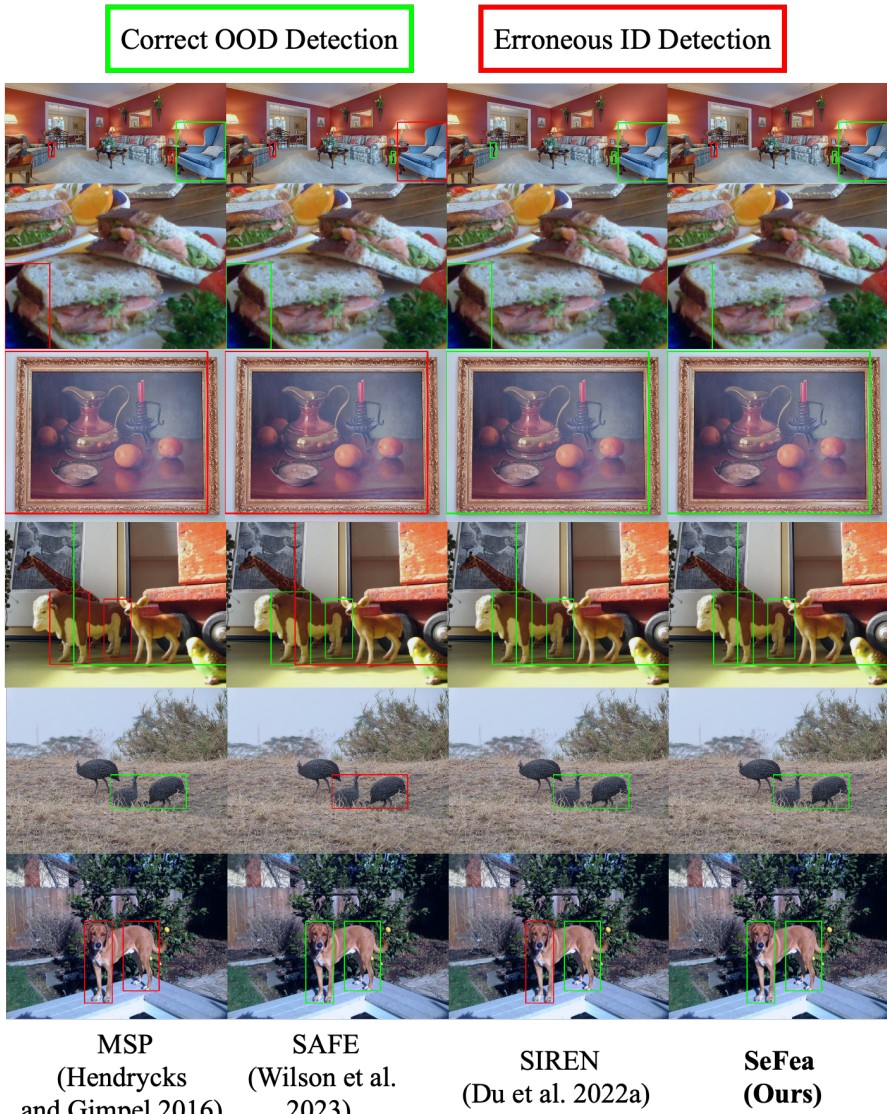

Correct OOD Detection    Erroneous ID Detection

| MSP (Hendrycks and Gimpel 2016) | SAFE (Wilson et al. 2023) | SIREN (Du et al. 2022a) | **SeFea (Ours)** |

Figure 5: Qualitative visualization of OOD-OD detection results from the MS-DETR object detector with **BDD as the ID dataset**, the images is collected from the COCO dataset. Detection results are obtained by comparing the OOD score against the threshold at FPR95. Green boxes indicate that the OOD detector correctly classifies the object as OOD, while red boxes denote misclassification as ID. All images are chosen such that none of the 10 BDD classes appear in the scenes.

## B ADDITIONAL RESULTS

### B.1 ADDITIONAL QUALITATIVE RESULTS

Additional visualizations are provided in Figures 4 and 5.

### B.2 INFERENCE TIME OVERHEAD

Table 4 illustrates the latency of the object detector and the additional overhead introduced by OOD detection, measured in FPS. We evaluate latency on two OOD datasets—COCO and OpenImages. Since different ID dataset configurations result in varying numbers of predicted bounding boxes, which in turn affect OOD-OD detection latency, we report results for both ID setups. The MS-

Table 4: Frames-per-second (FPS) metrics for the MS-DETR object detector and OOD-OD. Each cell reports the FPS for single-image inference. The table presents the latency of the OOD detector applied to each feature type, with SIREN-KNN used as the OOD detector for all feature types.

| | ID: PASCAL-VOC | | ID: BDD | | Avg FPS |
| | COCO | OpenImages | COCO | OpenImages | |
|---|---|---|---|---|---|
| Object Detector | 22 | 24 | 24 | 24 | 23 |
| OOD Detector – SAFE | 69 | 75 | 43 | 76 | 62 |
| OOD Detector – SIREN | 69 | 75 | 94 | 79 | **78** |
| OOD Detector – SeFea | 68 | 75 | 44 | 78 | 63 |

DETR object detector achieves approximately 23 FPS for object detection alone. SIREN-based feature OOD detection achieves the highest speed, as it is obtained solely from the penultimate layer. Interestingly, although both our approach and SAFE utilize four layers for OOD detection, our method achieves higher FPS for extraction because the selectively chosen OSFs in our approach produce smaller feature dimensionality than those used by SAFE. Notably, despite extracting features from four layers, our approach still attains a competitive FPS compared with the one-layer SIREN method (63 with 78).

Table 5: Ablation study on the number of top-$k$ sensitive layers used for OSFs collection in two transformer-based architectures: MS-DETR and ViTDET. All methods employ **SIREN-KNN** as the OOD detector. Evaluation is conducted on PASCAL-VOC and BDD as the ID datasets, with MS-COCO and OpenImages serving as the OOD datasets. Each reported value is the average over the two OOD datasets. Performance is measured using AUROC and FPR95. The bold cell indicates the best result, while the underlined cell indicates the second-best.

| | top-k sensitive layers | ID: PASCAL-VOC | | ID: BDD | |
| | | AUROC↑ | FPR95↓ | AUROC↑ | FPR95↓ |
|---|---|---|---|---|---|
| **MS-DETR** | 1 | 83.93 | 60.93 | 86.73 | 64.79 |
| | 2 | 86.41 | 53.70 | 88.56 | 65.46 |
| | 3 | **87.05** | 51.36 | 89.33 | 63.57 |
| | 4 | 86.30 | **50.42** | 89.56 | 60.97 |
| | 5 | 85.91 | 53.23 | **89.74** | **58.28** |
| | Penultimate | 82.57 | 56.42 | 89.18 | 63.01 |
| **ViTDET** | 1 | 90.46 | 43.85 | 91.24 | 43.02 |
| | 2 | 90.03 | 45.01 | **91.56** | 41.06 |
| | 3 | 89.18 | 45.06 | 91.47 | **40.15** |
| | 4 | **90.60** | **40.83** | 90.44 | 45.52 |
| | 5 | 89.08 | 44.62 | 89.82 | 46.80 |
| | Penultimate | 87.02 | 54.14 | 87.40 | 62.57 |

## B.3 K-SENSITIVE LAYERS

The effect of varying the number of top-$k$ sensitive layers for OOD detection is shown in Table 5. The AUROC score remains relatively stable across different numbers of $k$ sensitive layers. However, when $k = 4$, the results generally achieve well performance for both AUROC and FPR95 on MS-DETR and ViTDET, as indicated by the bold and underlined values representing the best and second-best results, respectively. Therefore, we select the top 4 sensitive layers for OSF collection, and all results for our proposed SeFea method in the main paper are reported using this $k$ value. In addition, the performance of the penultimate layer is also compared with that of the top-k layers, to explicitly demonstrate that the top-k layers outperform the penultimate layer for OOD detection.

## C  REPRODUCIBILITY DETAILS

### C.1  CODE

We have made the code publicly available under anonymity at the following GitHub link: GitHub Repository

### C.2  SENSITIVITY ANALYSIS - DETAILS

**Transformer-based models** MS-DETR (Wilson et al., 2023) is currently one of the SOTAs for object detecion task. It introduces additional one-to-many supervision for DETR-based methods, leading to several variants. In this paper, we adopt the variant of MS-DETR built upon Deformable DETR and refer to it simply as MS-DETR. The architecture consists of a CNN backbone, a transformer encoder, a transformer decoder, and prediction heads for object classes and bounding box positions. The encoder comprises six stacked transformer encoder blocks; each block contains SA layers and MLP layers that process feature maps from the CNN backbone. For analysis of SAFE features, we follow Wilson et al. (2023) and select batch norm / skip connection layers from CNN blocks.

ViTDET (Li et al., 2022) adapts the plain Vision Transformer for object detection by retaining a ViT backbone pre-trained with Masked Autoencoders (MAE), enabling strong single-image representations. In this paper, we focus on the ViT-B backbone, which consists of 12 Transformer blocks, 768-dimensional embeddings, and 12-head self-attention. During fine-tuning, ViTDET discards hierarchical backbones and classical feature pyramid networks (FPNs); instead, it constructs a lightweight four-level feature pyramid directly from the stride-16 output of the final ViT block, enabling multi-scale reasoning with minimal overhead. We investigate the distance-awareness properties of individual layers for out-of-distribution detection.

Table 6: As reference, we provide mAP on the ID datasets – VOC, and BDD – across different object detectors.

| Dataset | Method | mAP |
|---------|--------|-----|
| VOC | Deformable DETR (Du et al., 2022a) | 60.8 |
|     | MS-DETR (Zhao et al., 2024a) | 57.9 |
|     | ViTDET/ViT-B (Li et al., 2022) | 63.5 |
| BDD | Deformable DETR (Du et al., 2022a) | 31.3 |
|     | MS-DETR (Zhao et al., 2024a) | 33.1 |
|     | ViTDET/ViT-B (Li et al., 2022) | 34.9 |

**ID performance of Object Detectors** Since MS-DETR and ViTDET do not provide pretrained weights on the selected ID datasets, we train these models from scratch on the corresponding ID datasets. The ID performance, summarized in Table 6, confirms that the models are properly trained for evaluation of OOD-OD performance.

**Hyperparameters** We compute the Lipschitz norm by randomly sampling 5,000 pairs of bounding boxes. For $O_{l,d}$ in the input space ($x$), pooling along the spatial dimensions reduces the feature map to $1 \times 1$, yielding $p_{l,d}$ with only three channels. To increase dimensionality, for the input, we only pooling to $2 \times 4$ for VOC and $2 \times 2$ for BDD. The $2 \times 4$ size corresponds to the smallest bounding box dimensions in VOC, while for BDD-where the smallest bounding box is less than one pixel-we choose $2 \times 2$ to enhance dimensionality.

### C.3  EVALUATION METRICS

We assess OOD performance using two standard metrics-AUROC and FPR95-commonly adopted in prior OOD-OD studies (Liu et al., 2024; Wilson et al., 2023; Du et al., 2022b). AUROC measures the Area Under the Receiver Operating Characteristic Curve, which is calculated over multiple thresh-

olds; higher values indicate better performance, and 50% corresponds to random guessing. FPR95, on the other hand, reports the false positive rate when the true positive rate is at 95%, lower is better.

## C.4 COMPUTATIONAL RESOURCES

All experiments were conducted using Python 3.11.3 and PyTorch 2.3.0+cu121 on an NVIDIA RTX 6000 Ada Generation GPU (45 GB memory) running Ubuntu 22.04.3 LTS, equipped with an AMD Ryzen Threadripper PRO 5975WX 32-core processor. Please refer to the SAFE paper (Wilson et al., 2023) for environment installation instructions.

## D ALGORITHM *SeFea*

In Figure 1 of main paper we provide an overview of our OOD-OD method *SeFea*. Here we further provide the algorithm.

---

**Algorithm 1** Inference process of the proposed *SeFea* method for OOD-OD task.

---

**Input:** Input image $X$; object detector $f$; OOD score module $r$; a set of indices of the $k$ most sensitive layers $M$.
**Output:** Object-wise OOD predictions.

1. Perform inference using the object detector $f$ on input $X$ to obtain detected bounding boxes $\mathbb{B} = \{b_d\}_{d=1}^{D}$.

2. For each detected object $b_d$, extract the object-specific features (OSFs) from sensitive layers indexed in $M$, and concatenate them to form a unified representation:

$$p_d = \text{Concat}\left(\{p_{l,d}\}_{l \in M}\right).$$

3. Compute the OOD score for each object representation using the OOD score module $r$:

$$s_d = r(p_d).$$

---

As discussed in the main paper, sensitive transformer layers are determined by our proposed sensitivity metric $\text{sen}_{i,\cos}^{(\mathcal{T})}$, which has been validated to be correlated with OOD-OD accuracy. The indices of the sensitive transformer layers are stored in $M$, and the features of these layers are used by our proposed *SeFea* for OOD-OD task as in Algorithm 1. In our experiments, MS-DETR and ViTDET are explored as $f$, and SIREN-KNN and SIREN-vMF are explored as OOD score module $r$.

## E LLM USAGE

We used GPT-5 as a writing assistant for grammar checking and improving the formality of phrasing in sentences or short paragraphs. The model was not involved in research ideation, experimental design, analysis, or substantive content generation.

