# OpenReview forum: "Transformers Know When They Don’t Know: Layer-Wise Distance Awareness for OOD Detection"
_ICLR.cc/2026/Conference — Submitted to ICLR 2026_

### Official Review · Reviewer_RZRh · 2025-10-23

**Soundness:** 2
**Presentation:** 2
**Contribution:** 2
**Rating:** 2
**Confidence:** 4

**Summary:**

This paper investigates the task of OOD object detection. The authors investigate the role of feature embeddings from different layers of an object detector for this task, and proposing that features should be aggregated according to their “inut distance awareness”. They introduce SeFea, a post-hoc method that ranks layers by this metric, aggregates the features from the top-k layers and feeds them into a KNN/vMF-based OOD detector head (adapted from SIREN). They report improved performance over baselines on a common OOD object detection benchmark.

**Strengths:**

- The authors systematically test different transformation types (Table 2).
- That layers with highest sensitivity lead to high AUROC scores is also convincing (Figure 2).
- The proposed method is a post-hoc method and does not require retraining the object detector.

**Weaknesses:**

1. Baselines:
The authors only consider three baselines (MSP, SAFE, SIREN), and omit a large body of recent work [7,8,9,10,11,12,13]. They partly justify this claiming that those methods were developed for CNNs. However, in several cases, the methods simply use a CNN as OOD detection backbone (e.g. the Vista paper uses Faster RCNN), but I cannot see why their method could not be applied to a ViT out-of-the-box (similar for others). And even if this were not possible, it is worth mentioning that e.g. Vista achieves much lower FPR values (e.g. 8.68\% on BDD + OpenImages, vs 30.26\% in the best setting for SeFea). Further, the main baseline (SIREN [6]) is not used in its original form, as the authors state “although SIREN (Du et al., 2022a) retrains both the object detector and the OOD detector using a customized loss function, in our implementation, we only train the SIREN OOD detector while keeping the pretrained object detector frozen”.




2. Contribution and Novelty:
The proposed method is essentially an extension from Du et al [6], where the same OOD scoring rule is used. The difference is that this paper proposes aggregating different transformer layers, instead of using only the penultimate embeding layer. Using different layers for OOD detection is, however, not novel. The original Mahalanobis distance paper [1] used different layers, and there is also recent work on this, with more elaborate aggregation techniques [2,3,4]. The novelty thus stems from using a specific selection criterion for feature aggregation.



3. Benefits of aggregation are unclear:
There is no explicit comparison to other aggregation methods, and simple baselines like random selection, even selection across layers (e.g. one early, one intermediate, one late layer), last k layers, last layer + k-1 random layers, pooling all layers … Further doubts on the selection procedure are raised by 1) Figure 2 (left), where very small sensitivity scores yield the highest OOD detection performance and the penultimate layer is amongst the best-performing layers, and 2) Table 5, where for MS-DETR the most sensitive layer (k=1) is outperformed by the penultimate layer.


4. Overclaims:
I also think that at several points in the paper, the authors tend to overclaim their statements, and would advise to be more careful:
- “we are the first to exploit theoretical results of input distance awareness in Transformer architectures for OOD-OD” - SAFE also bases their selection of layers on the bi-lipschitz constraint
- “Code is in the Supp” There is no code (unless the authors refer to the pseudo code in Algorithm 1)
- “we achieve SOTA performance on challenging OOD-OD benchmarks”: like outlined above, other works report FPRs much lower than the present work
- “Unlike [...] methods that depend solely on penultimate-layer features  (Du et al., 2022a;b;
Kumar et al., 2023), our method can be applied without such architectural restrictions” While the mentioned references used the penultimate-layer features, their method is not “restricted” to this layer. So framing this as an inherent limitation of the method is misleading - in fact, the proposed method consists of applying the method from Du et al to more layers than only the last one.



[1] Kimin Lee, Kibok Lee, Honglak Lee, and Jinwoo Shin. A simple unified framework for detecting out-of-distribution samples and adversarial attacks. Advances in neural information processing systems, 31, 2018


[2] Colombo, P., Gomes, E. D., Staerman, G., Noiry, N., and Piantanida, P. Beyond mahalanobis-based scores for textual ood detection. arXiv preprint arXiv:2211.13527, 2022


[3] Rajasekaran, M., Sajol, M. S. I., Berglind, F., Mukhopadhyay, S., & Das, K. (2024). COMBOOD: A Semiparametric Approach for Detecting Out-of-distribution Data for Image Classification. In Proceedings of the 2024 SIAM International Conference on Data Mining (SDM) (pp. 643-651). https://doi.org/10.1137/1.9781611978032.74


[4] Tong Wei et al, harnessing Shallow Features in Pre-Trained Models for Out-of-Distribution Detection, https://openreview.net/forum?id=UTnq6hJJYa


[5] Samuel Wilson, Tobias Fischer, Feras Dayoub, Dimity Miller, and Niko S¨underhauf. Safe: Sensitivity-aware features for out-of-distribution object detection. In Proceedings of the IEEE/CVF international conference on computer vision, pp. 23565–23576, 2023


[6] Xuefeng Du, Gabriel Gozum, Yifei Ming, and Yixuan Li. Siren: Shaping representations for detecting out-of-distribution objects. Advances in Neural Information Processing Systems, 35:20434–20449, 2022


[7] Zhang et al, RUNA: Object-level Out-of-Distribution Detection via Regional Uncertainty Alignment of Multimodal Representations, AAAI 2025


[8] Zhang et al, VisTa: Visual-contextual and Text-augmented Zero-shot Object-level OOD Detection, ICASSP 2025


[9] Wu et al, Deep Feature Deblurring Diffusion for Detecting Out-of-Distribution Objects, ICCV 2023


[10] Isaac-Medina et al, Dream-Box: Object-wise Outlier Generation for Out-of-Distribution Detection, arxiv preprint: https://arxiv.org/pdf/2504.18746


[11] Liu et al, Can OOD Object detectors learn from foundation models, ECCV 2024


[12] Kumar et al, Normalizing Flow based Feature Synthesis for Outlier-Aware Object Detection, CVPR 2023


[13] Isaac-Medina et al, Towards Open-World Object-based Anomaly Detection via Self-Supervised Outlier Synthesis, arxiv preprint: https://arxiv.org/pdf/2407.15763

**Questions:**

In its current form, I do not see sufficient evidence for the effectiveness of the novel part of the paper (aggregating according to input distance awareness) compared to other aggregation methods and baselines. This would have to change significantly for me the change my score. In particular:
- Can the authors provide direct comparisons against the omitted methods [7-13]. If there is a reason not to do so, some justification would be required
- Can the authors provide ablation studies agains simpler aggregation baselines, e.g. a) random k layers, b) the last k layers, c) all layers pooled, d) evenly spaced layers, e) penultimate layer + random layers, … and the aggregations used in [2,3,4]. If not, why?

Minor remarks:
- the authors claim “Since these architectures are originally trained on the COCO dataset, we retrain them on the designated ID datasets prior to evaluation.” More details are required here. Is the retraining done from scratch? Or from the COCO checkpoint?
- The authors use the term “OOD accuracy” several times, and sometimes it seems to be confused with AUROC (e.g. in Figure 2)
- does the selection method also bring benefits for other detection methods (e.g. Mahalanobis, …)?

---

### Official Review · Reviewer_AfqC · 2025-10-27

**Soundness:** 3
**Presentation:** 2
**Contribution:** 3
**Rating:** 4
**Confidence:** 3

**Summary:**

The paper analyzes layer-wise input distance awareness in transformer object detectors and proposes SeFea. It is inspired from [1] that proposes a training recipe to encourage bi-Lipschitzness. In SeFea instead, authors rank layers by a sensitivity proxy (lower bound inspired bi-Lipschitz measure), using the object specific features from the top k layers. They score them using a SIREN-style head (either kNN or vMF). On MS-DETR and ViTDet with VOC/BDD as ID and using COCO/OpenImages as OOD datasets, SeFea improves AUROC/FPR95 over many benchmarks. The method does not change the used detector but trains an OOD head to perform OOD OD.

**Strengths:**

- **Transformer focus.** It moves the literature beyond CNN-centric OOD-OD and analyzes intermediate layers rather than defaulting to the default penultimate features.

- **Simple pipeline.** Sensitivity rank to multi-layer OSFs to lightweight OOD head while detector remains frozen is easy to follow and somewhat intuitive.

- **Correlational evidence.** The relationship between layer sensitivity is a good discovery although it comes with some caveats. I support the value of seeking for additional representational metrics to understand the network behavior to the ID and OOD samples.

- **Consistent gains on two detectors.** Achieves the best results on the evaluated benchmarks.

**Weaknesses:**

- **SOTA claims.** I believe OOD-OD area does not have yet a universally accepted and comprehensive benchmark (akin to OpenOOD [2] for classification). Claims would be more convincing if the method showed consistent gains across more detectors/architectures, training regimes and multiple OOD setups including near-OOD. (see the questions part for the actionable item)

- **Post-hoc vs training.** Although the detector stays frozen, the projection + prototype head is trained. To isolate the value of multi-layer OSFs, please add purely post-hoc baselines on OSFs (using kNN [3] or Mahalanobis++ [4] directly on the concatenated top-k OSFs. This would fit the story even better specifically because of the following point.

- **Lipschitzness claim vs trained head.** If the core premise is that input distance aware layers carry the right geometry, then adding a learned nonlinear detector on top of OSFs could distort that geometry when they are not regularized. Could the authors provide an intuition if they think the head will not break the distance awareness you rely.

- **Dimension invariant terminology and the metric design.** The phrase dimension invariant similarity measures (line 245) is confusing. The suggested metrics are meaningful only when both inputs live in the same vector space and they do not let you compare representations of different dimensionalities. Also, normalized Euclidean distance is a monotonic transformation of cosine similarity. Correct me if I am missing something but the difference in performances between normalized Euclidean distance vs cosine similarity should come from the extra $C_x / C_i$ scaling. In fact, $C_x / C_i$ makes the score dimension dependent, not dimension invariant on the other aspects. Please also report the results without $C_x / C_i$ to verify the intended behavior.

- **Pairing scheme.** The Gaussian pairing looks underspecified. The paper picks a mean/std but does not state how. Without a principled rule, the results are hard to interpret. For random pairs, I would like to see a controlled split into within class pairs, or weakly augmented pairs (e.g. using horizontal flip). Additional experiments could provide additional reasoning on the effect of pairing scheme.

- **k-layer aggregation.** Table 5 is not conclusive for an optimal k or whether the best k is stable across architectures or datasets. Without confidence intervals, specifically the gains beyond k=2 could be within noise.

- **Confidence intervals.** Reporting the mean +- standard deviation over 3 seeds for AUROC/FPR95 and also for FPS would make the results more convincing. I assume the training phase of the detector head is the main source of randomness. I would also appreciate if the authors can provide details on architecture choices and stopping criteria for the detector head.

**Questions:**

See my comments and questions below.

- **Using OpenOOD as a supporting argument.** Since the premise is about general layer sensitivity, adding a quick ImageNet-1k sanity check would strengthen the paper significantly. For standard pretrained ImageNet backbones (Swin/EVA/DeiT/ViT), rank layers by your sensitivity metric and compare pennultimate vs top-k intermediate layers and evaluate feature space OOD scores (kNN or Mahalanobis++ again) on the OpenOOD ImageNet suite. This will also include the near OOD cases and will stress test the claims.

- **Qualitative selection.** To avoid cherry-picking, either (i) randomly sample images with a fixed seed (provide it for reproducibility) or (ii) present per category montage grids built using pre-declared score thresholds. If unbiased comparative visuals are not feasible, focus the qualitative section on SeFea only examples with a discussion on failure cases.

- **Benchmark fairness.**  I am not sure whether the baseline results are obtained on the exact same backbones/checkpoints and with identical finetuning protocols. Could you confirm that this is the case?


## Minor comments

- Ablation and introduction lacks the "how" part. Readers understand that it is relevant to the input distance awareness but it would be more readable if the authors could explain the method briefly on a paragraph.

- Line114 misses an empty space.

- Line 142, stick either with "the retrained model" or "retrained model".

- Line 282 the sentence starting with "Neural networks..." requires citation.

- Line 287 I think it is hard to make robustness claims, I would temper the wording.

- Figure 2 has accuracy score as the axis name. I think it should be AUROC.

- Please refer to the specific section of the appendix/supplementary material when you refer to it on the main text.

- Please consider starting the appendix on a new page.

- For the FPS measure, it would be good to see the performance difference as you change k on a graph.


[1] Liu, J., Lin, Z., Padhy, S., Tran, D., Bedrax Weiss, T., & Lakshminarayanan, B. (2020). Simple and principled uncertainty estimation with deterministic deep learning via distance awareness. Advances in neural information processing systems, 33, 7498-7512.

[2] Zhang, J., Yang, J., Wang, P., Wang, H., Lin, Y., Zhang, H., ... & Li, H. (2023). Openood v1. 5: Enhanced benchmark for out-of-distribution detection. arXiv preprint arXiv:2306.09301.

[3] Sun, Y., Ming, Y., Zhu, X., & Li, Y. (2022, June). Out-of-distribution detection with deep nearest neighbors. In International conference on machine learning (pp. 20827-20840). PMLR.

[4] Mueller, M., & Hein, M. (2025). Mahalanobis++: Improving OOD Detection via Feature Normalization. arXiv preprint arXiv:2505.18032.

---

### Official Review · Reviewer_GuA7 · 2025-10-28

**Soundness:** 2
**Presentation:** 3
**Contribution:** 2
**Rating:** 4
**Confidence:** 4

**Summary:**

This paper addresses the problem of out-of-distribution object detection (OOD-OD). While transformer-based architectures have become dominant in object detection, prior research has mainly focused on OOD object synthesis or designing OOD detection scores, with little attention paid to the internal feature representations of transformers. To address this issue, the authors conduct the first in-depth analysis of transformer feature representations for OOD-OD. The authors develop a simple yet effective OOD detection method based on features from intermediate layers, achieving state-of-the-art performance across multiple OOD-OD benchmarks.

**Strengths:**

1. This paper is well-structured and easy to follow.

2. The analysis of internal layers within Transformer architectures for OOD-OD is interesting.

**Weaknesses:**

1. Since the idea of selectively incorporating intermediate representations to enhance OOD detection has been proposed in several prior works [1,2]. Could you please elaborate on how the proposed approach differs from and improves upon these existing works?

    [1] X-Mahalanobis: Transformer Feature Mixing for Reliable OOD Detection.

    [2] Mysteries of the Deep: Role of Intermediate Representations in Out of Distribution Detection.

2. Limited Scope of Comparison: The most recent OOD object detection methods are not compared [3, 4].

    [3] VOS: Learning what you don’t know by virtual outlier synthesis.

    [4] InfoBound: A Provable Information-Bounds Inspired Framework for Both OoD Generalization and OoD Detection

3. Does the proposed method influence the object detection ability on ID data?

4. In Lines 92-94, it is claimed that "we are the first to exploit theoretical results of input distance awareness in Transformer architectures for OOD-OD, leveraging Lipschitz analysis to quantify sensitivity". However, the paper does not appear to provide any formal theorems to validate the proposed method’s capability for OOD-OD.

5. From Table 2, Random Sampling is more effective in identifying informative intermediate layers for OOD detection. Does this result suggest that randomly sampled representations are inherently more OOD-like compared to those generated by FGSM or Gaussian perturbations? It would be helpful if the authors could provide further insights or analysis to explain this observation.

**Questions:**

See Weaknesses.

---

### Official Review · Reviewer_zEVJ · 2025-10-30

**Soundness:** 2
**Presentation:** 2
**Contribution:** 2
**Rating:** 4
**Confidence:** 3

**Summary:**

This paper presents a post-hoc way of detecting OOD samples for the object detection task. The proposed method is based on previous works of SAFE (sensitivity aware) and SIREN (OOD head and score). The paper evaluates several definitions of sensitivity and find the most relevant one. Based on the sensitivity score, features from the chosen layers are used to produce OOD score. The paper tested the method on two transformer based object detection models, and finds better performance than three baselines.

**Strengths:**

1. The study on the correlation between different sensitivity formulation and the AUROC is interesting. Although Tab. 2 shows that most realizations of the sensitivity have weak correlation (<0.5) with AUROC, which is against the motivation; The empirical study itself provides useful information for readers.
2. Results on Pascal-VOC and BDD show improved performance compared to baselines.

**Weaknesses:**

1. Missing important works [a, b] in the review of OOD detection for image classification, which studied OOD on vision transformers.
2. The presentation is sometimes vague and inexact. (1) L33. "the first study ... analyze transformer features for OOD detection" is over claimed. e.g., both [a, b] studied transformer features for OOD detection. (2) L824 "For analysis of SAFE features, we ... select ... layers from CNN blocks" Clearly this is not the SAFE paper. This text lacks careful edits. (3) L156 "... detecting OOD ... requires a deterministic model to be input distance-aware" This statement is speculating but is written like a fact. (4) L169 What is a "Gaussian process based output layer"? (5) It is unclear which layer is the "baseline layer" (penultimate of encoder/decoder/predictor/ood-detector?). (6) The y label of Fig. 2 should be AUROC instead of accuracy.
3. Lack ablation study. (a) The work highlights that the proposed method is "architecture-agnostic", yet it is only verified on two transformer based detectors. No CNN detectors are evaluated. (b) Only two transformer based detector is evaluated.
4. Typos: L112 emphase. Grammar issue: L113 Highlighting ...







- [a] Exploring the Limits of Out-of-Distribution Detection. NeurIPS 2021
- [b] ViM: Out-Of-Distribution with Virtual-logit Matching. CVPR 2022

**Questions:**

1. In Eq. (2), how is the Euclidean distance between inputs $x_j$ are computed? Are they computed on pixel values, or on patch embeddings?
2. The features from top-k sensitive layers are selected to compute the OOD score. To show the effectiveness of the particular selection criterior, it is better to compare with some naive way of layer selection, like (1) randomly select k layers, (2) select all attn layers, (3) select all mlp layers.
3. Why SAFE is not included in the VitDet section of Tab. 3?
4. Is the method applicable to more transformer based detectors, like DINO and MaskDINO?

---

### Meta-Review · Area_Chair_13Nq · 2025-12-31

**Summary:**

The paper investigates out-of-distribution object detection (OOD-OD) through the lens of transformer layer analysis. The authors propose "SeFea," a method that ranks and selects intermediate transformer layers based on a "sensitivity proxy" (input distance awareness) and aggregates their features for OOD scoring using a trained head.

While the reviewers acknowledged the importance of the topic and found the empirical study of layer-wise sensitivity interesting, the consensus is that the paper is not yet ready for publication. The primary reasons for the suggested rejection include:
1.  **Weak Baselines:** Multiple reviewers (RZRh, zEVJ, GuA7) pointed out that the paper compares against a very limited set of baselines and omits significant recent work in OOD-OD (e.g., VOS, Vista, RUNA).
2.  **Overclaimed Novelty:** The claim of being the "first" to analyze transformer features for OOD is challenged by existing literature (e.g., ViM, X-Mahalanobis).
3.  **Lack of Rigorous Ablation:** Reviewers requested comparisons against simpler aggregation strategies (e.g., random layer selection, pooling all layers) to justify the necessity of the proposed sensitivity-based ranking.
4.  **Methodological Clarity:** Concerns were raised regarding the "input distance awareness" theory versus the practice of training a non-linear head, which may distort the geometry the authors aim to preserve.

**Reviewer Concerns:**

The authors do not provide a rebuttal.

**Reviewer Scores:**

The authors do not provide a rebuttal.

---

### Decision · Program_Chairs · 2026-01-26

Reject